# SCoRe: Pre-Training for Context Representation in Conversational Semantic Parsing

**Tao Yu**[*]
Yale University
New Haven, CT, USA
`tao.yu@yale.edu`

**Rui Zhang**
Pennsylvania State University
State College, PA, USA
`r.zhang@psu.edu`

**Oleksandr Polozov, Christopher Meek & Ahmed Hassan Awadallah**
Microsoft Research
Redmond, WA, USA
`{polozov,meek,hassanam}@microsoft.com`

## Abstract

Conversational Semantic Parsing (CSP) is the task of converting a sequence of natural language queries to formal queries (e.g., SQL, SPARQL) to be executed against a structured ontology (e.g., databases, KBs). A CSP system needs to model the alignment between the unstructured language utterance and the structured ontology in the context of multi-turn dialog dynamics. Pre-trained language models have limited ability to represent NL references to structural data. We present SCoRe, a new pre-training approach for CSP tasks designed to induce representations that capture the alignment between the conversational flow and the structural context. By combining SCoRe with strong base systems on four different tasks (SParC, CoSQL, MWoZ, and SQA), we improve the performance over all baselines by a significant margin and achieve state-of-the-art results on three of them.

## 1 Introduction

Task-oriented dialog (TOD) systems (Tur & Mori, 2011) assist users in completing a task by performing an action or retrieving information backed by a structured ontology (e.g. a database, KB, or API). A key component of TOD is Conversational Semantic Parsing (CSP), which converts each dialog utterance into a formal query (e.g., SQL, SPARQL) to be executed against the ontology. CSP emerges in dialog systems (e.g., dialog state tracking in MWoZ (Budzianowski et al., 2018)), context-dependent semantic parsing (e.g., SParC (Yu et al., 2019b)), SQL-grounded state tracking (e.g., CoSQL (Yu et al., 2019a)), and sequential question answering (Iyyer et al., 2017). These settings differ, but all share the same objective and key challenge: *how to jointly represent the NL utterances and underlying structured ontology in the context of multi-turn dynamics of the dialog*.

CSP, like other NL tasks, benefits from pre-trained language models (PLMs) such as BERT (Devlin et al., 2019). However, general-purpose PLMs are pre-trained on free-form text using *language-driven* model objectives. This limits their ability in modeling the structured ontology context or the dialog dynamics. As a result, the emitted formal queries are often not grounded in the existing KB facts (Wang et al., 2020). In this work, we introduce SCoRe (**S**tructured & **S**equential **Co**ntext **Re**presentation), a language model pre-training approach for CSP. SCoRe adapts general PLMs by introducing a second phase of pre-training on synthesized CSP data with novel *context-driven* training objectives, which aim to ground utterances in the underlying ontology schema and model the conversational flow. SCoRe effectively injects structural and conversational inductive biases in PLMs that translate to many CSP tasks out-of-the-box. It does not require changes to the pre-trained model architecture and can be used as a drop-in replacement with any semantic parsing model.

We apply SCoRe to four different CSP tasks: (1) sequential text-to-SQL (SParC), (2) conversational text-to-SQL (CoSQL), (3) dialog state tracking (MWoZ), and (4) weakly-supervised sequential

---

[*]Work done during an internship at Microsoft Research.

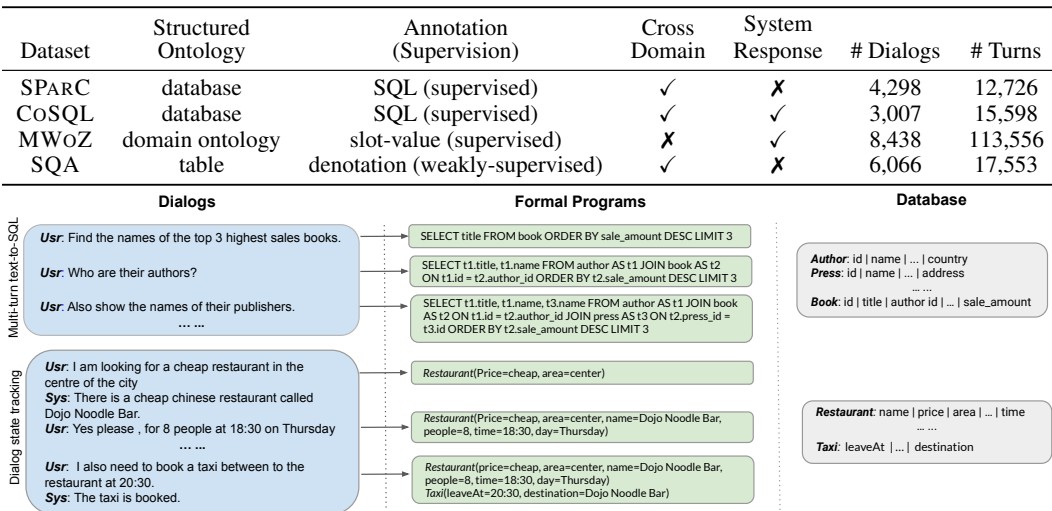

| Dataset | Structured Ontology | Annotation (Supervision) | Cross Domain | System Response | # Dialogs | # Turns |
|---|---|---|---|---|---|---|
| SPARC | database | SQL (supervised) | ✓ | ✗ | 4,298 | 12,726 |
| COSQL | database | SQL (supervised) | ✓ | ✓ | 3,007 | 15,598 |
| MWOZ | domain ontology | slot-value (supervised) | ✗ | ✓ | 8,438 | 113,556 |
| SQA | table | denotation (weakly-supervised) | ✓ | ✗ | 6,066 | 17,553 |

Figure 1: *Top:* Comparison of CSP datasets. *Bottom:* Two task examples from SPARC and MWOZ.

question answering (SQA). They represent different scenarios, ontologies, supervision signals, system responses, and domains (see Figure 1 for a detailed comparison and examples). We show that: (1) SCORE training objectives effectively incorporate synthetic data, (2) a single pre-trained SCORE model can be used for several CSP tasks and can be combined with different baseline systems and (3) SCORE significantly improves all baselines, achieves new state-of-the-art results on three benchmarks (SPARC, SPARC, and MWOZ) and comparable performance to state-of-the-art on SQA.

## 2 SCORE

The key challenge of CSP is to align the NL utterance and the structured ontology in the multi-turn dialog context. To this end, we inject structural and conversational inductive biases in SCORE using two novel objective functions: *Column Contextual Semantics (CCS)* and the *Turn Contextual Switch (TCS)*, in addition to established language-driven *Masked Language Modeling (MLM)*.

### 2.1 PRELIMINARIES

**Task Definition**  In CSP, at each turn $t$, we aim to produce a formal query $q_t$ given the current utterance $u_t$, the interaction history $h_t = [u_1, u_2, \ldots, u_{t-1}]$, and the schema $c$ (table and column names, slots, etc.) of the target database (ontology) $d$. The tasks we consider (Figure 1) differ in their target formal language and ontology:

- The **utterance** $u$ is the user question for SPARC and SQA, while for COSQL and MWOZ, $u$ is the combination of a user query and a system response.
- The **database** $d$ is used verbatim (multi-table) for SPARC and COSQL; for MWOZ, the pre-defined ontology $d$ can be viewed as a database; for SQA, $d$ is a single table.
- The **formal query** $q$ is SQL for SPARC and COSQL; for MWOZ it is the slot-value pairs, viewed as simple SQL consisting of SELECT and WHERE; and for SQA, $q$ is the latent program.

**Base Architecture**  The base architecture of SCORE takes as input a single turn of a CSP dialog $C_t = \langle u_t, h_t, c \rangle$, and encodes it into *contextualized conversation representations* $\vec{S}_t$ for each token in $C_t$. The encoder architecture follows RoBERTa (Liu et al., 2019). It is then followed by a linear layer and normalized (Ba et al., 2016) to produce final representations $\vec{h}_t$ for each token:

$$C_t = \langle u_t, v_t, c \rangle, \quad \vec{S}_t = \text{SCORE}(C_t), \quad h_{t,i} = \text{LayerNorm}(\text{GELU}(W_1 S_{t,i})) \, \forall \, S_{t,i} \in \vec{S}_t, \quad (1)$$

where GELU is an activation by Hendrycks & Gimpel (2016) and $W_1$ is a learned parameter matrix. The *context representations* $\vec{h}_t$ are then used as input to a *program decoder model* to produce the formal query $q_t = f_{\text{dec}}(\vec{h}_t \mid C_t)$. As §3 details, we use different state-of-the-art models for $f_{\text{dec}}$.

## 2.2 Pre-training Objectives

**Column Contextual Semantics**   To address alignment between the utterance and the underlying database schema, we optimize SCORE with the auxiliary objective of *Column Contextual Semantics (CCS)*. For each column in the schema, CCS targets the *operations* that should be performed on it in a given dialog turn. Each query $q$ is decomposed into operations on columns and tables, e.g. GROUPBY and HAVING for SQL queries, or WHERE for slot-value pairs. The CCS loss is given by:

$$\mathcal{L}_{\text{CCS}}(C_t) = \sum_i \text{CrossEntropy}_{148}(\text{LayerNorm}(\boldsymbol{W}_2\,\boldsymbol{h}^c_{t,i})) \tag{2}$$

where $\boldsymbol{h}^c_{t,i}$ is the contextualized representation of the $i^{\text{th}}$ column's initial special token in $C_t$.

**Turn Contextual Switch**   The *Turn Contextual Switch (TCS)* objective targets predicting *the difference in formal queries between dialog turns* based on the utterance to capture the conversational context flow. Using the context-free grammar of SQL, we identify 26 possible *kinds of differences* that a conversational turn could elicit. For example, DEL(SELECT.agg) indicates removing an aggregate operation from a SELECT statement (e.g. when an utterance *"Show all the ages instead."* elicits a change SELECT MAX(age) ... → SELECT age ...). The TCS loss targets predicting the context switch label between each conversational turn and the previous history:

$$\mathcal{L}_{\text{TCS}}(C_t) = \text{CrossEntropy}_{26}(\text{LayerNorm}(\boldsymbol{W}_3\boldsymbol{H}^s_t)) \tag{3}$$

where $\boldsymbol{H}^s_t \in \mathbb{R}^{(t-1)\times d}$ is the contextualized representation of all previous turns in $C_t$ with hidden dimension $d$. TCS is not used to pre-train on MWOZ because its context switch labels are trivial.

**Masked Language Modeling**   As in prior work on large-scale language models (Devlin et al., 2019), we use the *Masked Language Modeling (MLM)* objective to facilitate contextual representation learning for natural language utterances. Importantly for regularization, we only apply this loss on *in-domain human-annotated* NL data – namely, SPARC, COSQL, SQA, and nine TOD datasets by Wu et al. (2020). Formally, on each pre-training step we mask 15% of tokens in $C_t$ and optimize

$$\mathcal{L}_{\text{MLM}}(C_t) = \sum_m \text{CrossEntropy}_{\text{Vocab}}(\text{LayerNorm}(\boldsymbol{W}_4\boldsymbol{h}^m_t)) \tag{4}$$

where $\boldsymbol{h}^m_t$ are the contextualized representations of the masked tokens in $C_t$.

**Pre-Training Setup**   As Table 1 indicates, the total number of labeled dialog turns in CSP datasets is ca. 160,000. Following established practice, we augment them with *synthesized conversational data*. We incorporate it into pre-training, forming an auxiliary dataset $\mathcal{D}_{\text{syn}}$ in addition to the naturally supervised dataset $\mathcal{D}_{\text{sup}}$. The total pre-training loss is the sum of the three with CCS and TCS only applied to $\mathcal{D}_{\text{syn}}$ and MLM only to $\mathcal{D}_{\text{sup}}$:

$$\mathcal{L} = \sum_{C_t \in \mathcal{D}_{\text{syn}}}\left(\mathcal{L}_{\text{CCS}}(C_t) + \mathcal{L}_{\text{TCS}}(C_t)\right) + \sum_{C_t \in \mathcal{D}_{\text{sup}}}\mathcal{L}_{\text{MLM}}(C_t) \tag{5}$$

We re-use the dataset of 120k synthetic task-oriented dialogues for MWOZ, introduced by Campagna et al. (2020). In this work, we introduce a complementary procedure to synthesize data for *conversational* text-to-SQL dialogues. We use about 400k tables in WIKITABLES (Bhagavatula et al., 2015), WikiSQL (Zhong et al., 2017), and Spider (Yu et al., 2018b) datasets as underlying databases $d$, over which we generate 435k text-to-SQL conversations.

To this end, we induce two context-free utterance-SQL generation grammars: (1) a single-turn grammar $G_s$ for generating context-independent question-SQL pairs, and (2) a follow-up grammar $G_c$ for conversational question-SQL pairs. The follow-up grammar $G_c$ contains context switch labels and lists of follow-up question templates, e.g. a label INS(SELECT.column0) could generate the question *"How about show column0 too?"* To ensure generalization, we only induce the grammars from the SPARC training set. Appendix B shows examples of the templates and synthetic utterances.

## 3  Experiments

**Metrics & Base Models**   We evaluate SCORE on four CSP tasks in Figure 1, using the official metrics for each. For SPARC and COSQL, we report *question match accuracy (QM)*, the exact set

| | SPARC | | | | COSQL | | | |
|---|---|---|---|---|---|---|---|---|
| | Dev | | Test | | Dev | | Test | |
| Models | QM | IM | QM | IM | QM | IM | QM | IM |
| SyntaxSQL (Yu et al., 2018a) | 18.5 | 4.3 | 20.2 | 5.2 | - | - | 14.2 | 2.2 |
| GAZP + BERT (Zhong et al., 2020) | 48.9 | 29.7 | 45.9 | 23.5 | 42.0 | 12.3 | 39.7 | 12.8 |
| EditSQL + BERT (Zhang et al., 2019) | 47.2 | 29.5 | 47.9 | 25.3 | 39.9 | 12.3 | 40.8 | 13.7 |
| IGSQL + BERT | 50.7 | 32.5 | 51.2 | 29.5 | 44.1 | 15.8 | 42.5 | 15.0 |
| $R^2$SQL + BERT | - | - | 55.8 | 30.8 | - | - | 46.8 | 17.0 |
| RAT-SQL + BERT (Wang et al., 2019) | 56.8 | 33.4 | - | - | 48.4 | 19.1 | - | - |
| + ROBERTA | 58.2 | 36.7 | - | - | 50.1 | 19.3 | - | - |
| + SCORE | **62.2** | **42.5** | **62.4** | **38.1** | **52.1** | **22.0** | **51.6** | **21.2** |

Table 1: The SPARC and COSQL accuracy over all questions (QM) and all interactions (IM). The scores of IGSQL + BERT and $R^2$SQL + BERT are from the official leaderboards.

| Models | MultiWOZ 2.1 |
|---|---|
| TRADE | 46.60 |
| DS-DST | 51.21 |
| SOM-DST | 52.57 |
| DS-picklist | 53.30 |
| TripPy | 55.29 |
| SimpleToD | 55.72 |
| TripPy (ours) | 58.37 |
| + SCORE | **60.48** |

Table 2: Joint goal accuracies (JGA) on the MWOZ 2.1 test set.

| | SQA | |
|---|---|---|
| Models | QM | IM |
| Iyyer et al. (2017) | 44.7 | 12.8 |
| Sun et al. (2019) | 45.6 | 13.2 |
| Müller et al. (2019) | 55.1 | 28.1 |
| Herzig et al. (2020) | **67.2** | **40.4** |
| Wang et al. (2019) + RoBERTa | 62.8 | 33.2 |
| with 10% training data | 53.3 | 21.2 |
| Wang et al. (2019) + SCORE | 65.4 | 38.1 |
| with 10% training data | 57.1 | 26.1 |

Table 3: QM and IM accuracy on the SQA test set.

match accuracy over SQL templates (Yu et al., 2018b), and *interaction match accuracy (IM)*, the ratio of interactions with all questions predicted correctly. For MWOZ, we report joint goal accuracy (JGA) – similar to the IM accuracy used in SPARC and COSQL. For SQA, we report denotation QM and IM accuracies.

For **SPARC and COSQL**, we use RAT-SQL (Wang et al., 2020) as our base model. Since it is originally developed for single-turn text-to-SQL, we extend it to a multi-turn setting by concatenating current utterances and dialog history. For **MWOZ**, we use the state-of-the-art Trippy model (Heck et al., 2020), which uses $BERT_{base}$ to encode the utterances and dialog history. We report $\sim 2\%$ higher results because we train it for more epochs (25 vs. 10). For **SQA**, we use the semantic parser by Wang et al. (2019), extended to multi-turn similarly to RAT-SQL. It first generates a program template and then instantiates it by searching for matches between template slots and question spans.

**Overall Results**   The results of SPARC and COSQL, MWOZ, and SQA are in Table 1, 2, and 3 respectively. We run each main experiment three times with different random seeds and report the mean. Overall, SCORE gains significant improvements over BERT and ROBERTA on all tasks, achieving state-of-the-art performances on SPARC, COSQL, and MWOZ. We also show that SCORE outperforms ROBERTA under a few-shot setting of SQA when only 10% of training data is available.

**What is the effect of each pre-training objective?**   Table 4 shows an ablation study on different pre-training objectives. We find that the performance drops for COSQL and MWOZ while increases for SPARC when removing the MLM loss (CCS+TCS). One possible reason is that SPARC is created by

| | SPARC | COSQL | MWOZ | SQA |
|---|---|---|---|---|
| CCS + TCS + MLM | 38.6 | 21.7 | 60.48 | 33.7 |
| MLM only | 37.0 | 20.3 | 59.47 | 34.7 |
| CCS only | 41.3 | 21.2 | 59.32 | 32.7 |
| CCS + TCS | 42.5 | 22.0 | - | 38.1 |

Table 4: The effect of SCORE pre-training objectives.

decomposing complex questions in sequences of inner related simpler questions with more strict underlying patterns and less language diversity. Also, the synthesized data used to pre-train SCORE for SPARC and COSQL is generated by the grammar induced by SPARC, which might overfit to SPARC. In addition, SCORE pre-trained with only MLM loss doesn't help but even hurt the performance on SPARC. For the other tasks, the MLM loss slightly improves the performance especially

on COSQL. Finally, we test the effectiveness of TCS on SPARC and COSQL (CCS only vs. CCS + TCS), SCORE gains improvements of 1.5% on SPARC and 0.5% on COSQL.

**What if we directly augment the training set with synthetic data?** We compare the base models trained with or without synthetic data on COSQL and MWOZ. As shown in Table 5, synthetic data augmentation in the training set does not significantly improve performance, confirming similar findings in recent works (Zhang et al., 2019; Herzig et al., 2020; Campagna et al., 2020; Zhong et al., 2020). In contrast, pre-training on it using our objectives yields downstream improvements.

|  | COSQL | MWOZ |
|---|---|---|
| no syn | 48.4 | 58.37 |
| with syn | 48.6 | 58.45 |

Table 5: Effect of synthetic data as training data augmentation.

## 4 CONCLUSION

Conversational semantic parsing is one of the most important research topics in conversational AI and has been studied in different settings including task-oriented dialogue (Budzianowski et al., 2018), question answering (Iyyer et al., 2017), and text-to-SQL (Yu et al., 2019a;b). We present SCORE, a new pre-training approach for CSP. The training objectives of SCORE aim to induce natural language representations that capture the multi-turn dynamics, compositional semantic of the target language, and the references to the structural ontology appearing in the dialog. We demonstrated SCORE effectiveness by using it as a drop-in feature representation encoder with strong baseline models for four different CSP tasks and achieving state-of-the-art results on three of them.

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

## A   DETAILED RESULTS

**Does SCORE improve question match accuracy on individual turns?**   Table 6 shows detailed results of SCORE's question accuracy for individual conversation turns on the SPARC dev set. SCORE provides a significant improvement for every conversation turn except the first (in which the task is more similar to single-turn semantic parsing). COSQL and SQA exhibit similar behavior, as shown in Table 7 and Table 8.

|  | QM | Q1 | Q2 | Q3 | Q4 |
|---|---|---|---|---|---|
| RAT-SQL + BERT | 56.8 | **71.1** | 53.6 | 47.8 | 31.8 |
| +RoBERTa | 58.2 | 68.7 | 58.5 | 48.9 | 35.2 |
| + SCORE | **62.2** | 70.6 | **63.5** | **52.6** | **45.5** |

Table 6: Detailed results on the dev set of SPARC. $Q_i$ is the accuracy of the $i^{\text{th}}$ conversation question.

|  | QM | IM | Q1 | Q2 | Q3 | Q4 | Q5 |
|---|---|---|---|---|---|---|---|
| RAT-SQL + BERT | 48.4 | 19.1 | 54.6 | 48.4 | **47.5** | 43.9 | 31.0 |
| +RoBERTa | 50.1 | 19.3 | 59.7 | 50.9 | 46.3 | 46.5 | **32.4** |
| + SCORE | **52.1** | **22.0** | **60.8** | **53.0** | **47.5** | **49.1** | **32.4** |

Table 7: Detailed results of COSQL on the dev set.

|  | QM | IM | Q1 | Q2 | Q3 |
|---|---|---|---|---|---|
| Wang et al. (2019) | 51.0 | 22.0 | 68.3 | 48.0 | 38.5 |
| +RoBERTa | 62.8 | 33.2 | 77.2 | 61.7 | 52.1 |
| +SCORE | **65.4** | **38.1** | **78.3** | **65.3** | **54.9** |
| Few-Shot (10% training data) | | | | | |
| Wang et al. (2019) | | | | | |
| +RoBERTa | 53.3 | 21.2 | 71.0 | 52.5 | 36.6 |
| +SCORE | **57.1** | **26.7** | **74.6** | **56.7** | **40.7** |

Table 8: Detailed results of SQA on the test set.

**How general is SCORE?**   In addition to generalization over different *task settings*, we demonstrate the effectiveness of SCORE on different *base models*. To this end, we experiment with a different base model SOM-DST for MWOZ. As shown in Table 9, SCORE can still improve the performance with a different base model on MWOZ (SOM-DST+BERT vs. SOM-DST+SCORE on syn. MWOZ).

|  | MWOZ |
|---|---|
| SOM-DST + BERT | 52.57 |
| + SCORE on syn. text-to-SQL | 53.57 |
| + SCORE on syn. MWOZ | 54.61 |

Table 9: Performance of SCORE pre-trained on different synthesized data on MWOZ.

For generalization in synthetic grammar and data, as shown in Table 1 and 3, SCORE is pre-trained on the synthesized data of the grammar induced from SPARC, and it still improves the performance on COSQL and SQA. Moreover, in Table 9 we show that SCORE pre-trained on the text-to-SQL synthesized data could also surprisingly improve the performance on MWOZ. We expect that higher performance could be achieved with SCORE pre-trained on task-specific synthesized data.

## B   SYNTHESIZED EXAMPLES & TEMPLATES

Table 10 shows an example of the synthesized question-SQL pairs and their corresponding templates in our grammars.

| Turn # | Question-SQL Template | Synthesized Question-SQL |
|---|---|---|
| 1 | "Find the number of TABLE0 with COLUMN0 OP0 VALUE0" SELECT COUNT(*) ORDER BY COLUMN0 OP0 VALUE0 | "Find the number of football team with team hometown is not murrieta, california?" SELECT COUNT(*) WHERE TEAM_HOMETOWN != "MURRIETA, CALIFORNIA" |
| 2 | "Can you give me their COLUMN1?" TCS:       REPLACE(SELECT.COLUMN0), DEL(SELECT.AGG) | "Can you give me their football team player?" SELECT FOOTBALL_TEAM_PLAYER WHERE TEAM_HOMETOWN != "MURRIETA, CALIFORNIA" |
| 3 | "How about only show those with AS0 COLUMN2?" TCS: ADD(ORDERBY_AS0.COLUMN2) | "How about only show those with the largest age?" SELECT FOOTBALL_TEAM_PLAYER WHERE TEAM_HOMETOWN != "MURRIETA, CALIFORNIA" ORDER BY AGE DESC LIMIT 1 |
| 4 | "AS1?" TCS: REPLACE(ORDERBY_AS1.COLUMN2) | "The smallest?" SELECT FOOTBALL_TEAM_PLAYER WHERE TEAM_HOMETOWN != "MURRIETA, CALIFORNIA" ORDER BY AGE AS LIMIT 1 |

Table 10: An example of synthetic conversational text-to-SQL data.

## C  PRE-TRAINING COST

We test the performance of SCORE with respect to the number of pre-training epochs. Figure 2 shows that the best performance of the downstream tasks is usually achieved in early epochs, more specifically 5 for SPARC and COSQL and 15 for MWOZ. Longer pre-training time does not improve or even hurts the performance. One possible reason is that longer pre-training makes SCORE overfit to the synthesized data whose utterances are unnatural.

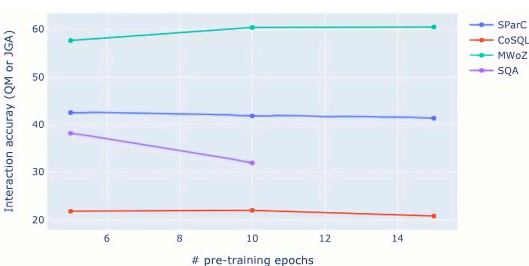

Figure 2: The effect of pre-training time.

As for the data, as shown in Table 4, even if SCORE is pre-trained with only a relatively small amount of synthesized data (without the MLM loss), most of the tasks can achieve much higher performances. With a relatively smaller training corpus and shorter training time compared to other pre-trained language models, SCORE is efficient in time and data.

## D  IMPLEMENTATION DETAILS

### D.1  SCORE

For pre-training SCORE on synthesized text-to-SQL data, we use ROBERTA $_{large}$ and pre-train it with batch size 12, gradient accumulation step 2, and maximum length 248. We use a learning rate $1e$-5 and gradually reduce the learning rate without a warm-up period using Adam (Kingma & Ba, 2014) with epsilon $1e$-8. BERT$_{base}$ is used in pre-training SCORE on synthesized MWOZ data because it contains longer conversations. We set the maximum length to 512 and batch size 24. All SCORE are pre-trained for 30 epochs, which usually take less than half a day on 8 V100 GPUs. We experimented with SCORE pre-trained for 5, 10, and 30 epochs and found that most of the best downstream performances occur when base systems incorporate with SCORE pre-trained for less than 10 epochs. Our implementation is based on the Transformers library (Wolf et al., 2019).

### D.2  BASE MODELS

**RAT-SQL:** For a fair comparison, all RAT-SQL experiments are trained for 40k steps. We adopt the same hyperparameters as Shaw et al. (2018) except for learning rates. We find that learning rates of $1e$-4 and $1e$-5 for RAT and BERT respectively produce more stable results.

**TripPy:** We use the same hyperparameters for training TripPy on MWOZ as in (Heck et al., 2020) except we train it for 25 epochs (as opposed to 10 epochs as reported in (Heck et al., 2020)). When

we train TripPy for 25 epochs, we get a new result that is higher (around 2%) than the one reported in (Heck et al., 2020). Similarly, when we train TripPy with SCORE, we train it for 25 epochs.

**SOM-DST:** We use the same hyperparameters from Kim et al. (2020) for all SOM-DST experiments on MWOZ.

**Wang et al. (2019):** We use the same hyperparameters from Wang et al. (2019) for SQA experiments.

## E  TASK-ORIENTED DIALOGUE DATASETS

| Name | # Dialogue | # Utterance | Avg. Turn | # Domain |
|---|---|---|---|---|
| MetaLWOZ (Lee et al., 2019) | 37,884 | 432,036 | 11.4 | 47 |
| Schema (Rastogi et al., 2019) | 22,825 | 463,284 | 20.3 | 17 |
| Taskmaster (Byrne et al., 2019) | 13,215 | 303,066 | 22.9 | 6 |
| MWOZ (Budzianowski et al., 2018) | 10,420 | 71,410 | 6.9 | 7 |
| MSR-E2E (Li et al., 2018) | 10,087 | 74,686 | 7.4 | 3 |
| SMD (Eric and Manning, 2017) | 3,031 | 15,928 | 5.3 | 3 |
| Frames (Asri et al., 2017) | 1,369 | 19,986 | 14.6 | 3 |
| WOZ (Mrkšić et al., 2016) | 1,200 | 5,012 | 4.2 | 1 |
| CamRest676 (Wen et al., 2016) | 676 | 2,744 | 4.1 | 1 |

Figure 3: Data statistics of human-annotated task-oriented dialogue datasets used in Wu et al. (2020).

