# OpenReview forum: "SCoRe: Pre-Training for Context Representation in Conversational Semantic Parsing"
_NeurIPS.cc/2020/Workshop/CAP — NeurIPS 2020 CAP Workshop_

### Official Review · AnonReviewer1 · 2020-10-22
**a practical set of tools and tricks for making pre-training for semantics parsing work better**

**Rating:** 6
**Confidence:** 3

**Review:**

I think the paper is solid, and the results look convincing that these tricks help in training the semantic parser by giving the training procedure more context of the underlying programmatic semantics during training. in a sense this is similar to the recent works on execution-guided program synthesis.

My "complaint" about this work is that there just seems to be a lack of a central "theme" of what they're doing. The results are good, but if my domain of semantic parsing is slightly different than getting a system to translate from NL to a database query, would some of the same principles of this paper still apply? Between the many 3-lettered acronyms I felt the central message is kinda lost.

So some re-structuring and distillation of the key idea would be great, the results look good.

---

### Decision · Program_Chairs · 2020-11-02

**Decision:**

Accept

**Comment:**

The review is sufficiently positive that I am recommending acceptance.